# Newly Woody Artificial Diet Reveals Antibacterial Activity of Hemolymph in Larvae of *Zophobas atratus* (Fabricius, 1775) (Coleoptera: Tenebrionidae)

**DOI:** 10.3390/insects15060435

**Published:** 2024-06-08

**Authors:** Alexander Kuprin, Vladislava Baklanova, Maria Khandy, Andrei Grinchenko, Vadim Kumeiko

**Affiliations:** 1Federal Scientific Center of the East Asia Terrestrial Biodiversity, Far East Branch, Russian Academy of Sciences, Vladivostok 690022, Russia; 2School of Medicine and Life Sciences, Far Eastern Federal University, Russky Island, Vladivostok 690091, Russia; 3A.V. Zhirmunsky National Scientific Center of Marine Biology, Far East Branch, Russian Academy of Sciences, Vladivostok 690041, Russia

**Keywords:** saproxylic beetles, mass rearing, diet, development, antibacterial activity, hemolymph, Coleoptera, *Zophobas atratus* (Fabricius, 1775)

## Abstract

**Simple Summary:**

Currently, the optimization of mass rearing conditions for insects is a matter of highly applied and fundamental relevance since insects can be used as a source of protein in feed and feed supplements for farm animals as well as a source of antibacterial compounds for drug development and various insect research activities in laboratory conditions. The quantity and quality of the consumed feed has an effect on insect growth rate, duration of development, body weight, ability to spread, and survival, as well as on the mating success rate and total egg production of females. Suboptimal nutrition, growth, and development can affect beetle viability. We developed an artificial fungi-based diet for rearing an insect group adapted to life in wood and using it as one of their food sources and tested the effect of this diet on various parameters of the insects’ development and their immunity. *Zophobas atratus* (Fabricius, 1775) (Coleoptera: Tenebrionidae) was chosen as a model species (representative of this group of insects). Based on the parameters of development at all stages of *Z. atratus* and screening of hemolymph antibacterial activity of their larvae, we showed that the cultivation method we developed shortens the development time of beetles and strengthens their immunity.

**Abstract:**

The rearing of saproxylic insects in laboratory conditions is an important task for studying the biology of insects. Through understanding nutritional needs, it is possible to optimize beetle rearing in laboratory conditions. In this study, an artificial fungi-based diet (FD) was developed for the cultivation of the darkling beetle *Zophobas atratus* (Fabricius, 1775) (Coleoptera: Tenebrionidae) in laboratory conditions as a model object for studying the biology of saproxylophagous beetles. To assess the influence of the diet, a number of physiological parameters were measured, including development time, body size, and weight of all stages of the beetle’s life cycle, as well as its immune status. The immune status of *Z. atratus* was assessed on the basis of larval hemolymph antibacterial activity against six different bacterial strains assessed using disk-diffusion and photometric tests. Our findings show that the FD reduces development time and boosts the immune status as compared to beetles reared on a standard diet (SD). Samples from FD-reared larvae had pronounced antibacterial activity as compared to samples from SD-reared larvae. This work is of fundamental importance for understanding the correlations between nutrition and development of saproxylic Coleoptera and is the first report on immune status regulation in this group of insects.

## 1. Introduction

Model organisms are often used in biological, medical, and environmental research. The list of model organisms among insects is limited to very few, including the fruit fly *Drosophila melanogaster* Meigen, 1830 (Diptera: Drosophilidae), *Galleria mellonella* (Linnaeus, 1758) (Lepidoptera: Pyralidae), *Bombyx mori* Linnaeus, 1758 (Lepidoptera: Bombycidae), *Periplaneta americana* (Linnaeus, 1758) (Blattodea: Blattidae), and *Locusta migratoria* Linnaeus, 1758 (Orthoptera: Acrididae) [1,2,3,4]. In addition to these insect species, an increasing number of insect species have recently been proposed as model organisms due to their worldwide distribution and ecological importance, the possibility of extrapolating studies to vertebrates, and the relatively low cost of culturing [5,6,7]. The last decade has seen growing interest in the Coleoptera order, which is the richest order in the animal kingdom, numbering around 400,000 species that make up almost 25% of all known animal life forms and have occupied almost all terrestrial habitats [8]. Beetles interact with plants and other organisms, as well as with dead and decomposing materials, thus playing a key role in natural and man-made ecosystems [9].

A challenge in dealing with Coleoptera species is the continuous monitoring in their natural habitat (in soil and inside tree trunks). That is why their rearing in artificially created environments significantly simplifies the task. By changing the cultivation conditions, one can influence various functions at all stages of insect development. In this regard, the optimization of conditions and development of various substrates for the mass rearing of insects are believed to be relevant fundamental and applied lines of research in entomology, and the corresponding scientific results can be used in various areas of biotechnology. To date, considerable material has been accumulated on the influence of various environmental parameters and diet composition on the development of many insect groups in laboratory conditions [10,11,12,13]. Undoubtedly, nutrition is the main factor influencing growth and development, as well as the accumulation of insect mass [14,15,16,17]. However, little is known about the impact that feed quality has on correlated immunological functions. This question is crucial for understanding insect adaptation strategies to specific environmental conditions.

Immunity is an important internal factor affecting the viability of insects [18,19,20,21,22]. The immune system of insects is based mainly on innate immune mechanisms, i.e., cellular and humoral responses. The cellular response includes all processes involving hemocytes, the cellular components of insect hemolymph [23]. The humoral response includes processes involving mainly molecules such as antimicrobial peptides. It has been established that the main source of plasma proteins including antimicrobial peptides (AMPs) is the fat body [24,25]. It creates effective concentrations of bactericidal molecules in the hemolymph, which provides information on the physiological state of insects [26]. The immune system of saproxylic Coleoptera, as highly developed insects, is believed to be well developed. Moreover, they live in decaying wood and debris and need to control the active growth of pathogenic bacteria and other organisms that are abundant there [27,28,29].

The *Zophobas atratus* (Fabricius, 1775) (Coleoptera: Tenebrionidae) beetle from the family of darkling beetles lives in tropical regions of Central and South America. Its larvae are saproxylophagous and are trophically associated with large woody debris of deciduous trees; there is also evidence of them feeding on bat guano [30]. Currently, this species has been introduced to many regions of Europe and Asia. Its high fecundity and simple management make it a convenient model species for various experiments in research on insect biology and insect adaptations under the influence of various environmental factors, as well as molecular genetic and biotechnological research [31,32,33,34].

The objective of this work was to study the effect of the feed composition on the antibacterial activity of the hemolymph of *Z. atratus* larvae cultivated in laboratory conditions.

## 2. Materials and Methods

### 2.1. Stock Culture

The stock culture (10 pairs of adult beetles) was purchased at a pet shop (Vladivostok, Primorsky Krai, Russia). The acquired 10 pairs of adult beetles were kept in laboratory conditions in 5-litre plastic cages with 5 cm of sawdust of Japanese elm and leaf litter (to maintain moisture) on the bottom of the cage as well as large branches with bark of the fourth decay class to make space for the beetles to create shelters and lay eggs. Various sorts of fruit puree, fresh fruit, and, rarely, vegetables were used as an additional source of food. Two to three times a week, the breeding cell was sprayed with filtered or distilled water to maintain humidity within 60 and 75%. The laid eggs were transferred to cages with two different substrates, as described in Section 2.2.

### 2.2. Cultivation of Z. atratus Larvae Biomass with Various Diets

The first substrate was a fungi-based diet (FD), which corresponds to the trophical preferences of darkling beetles and their saproxylophagous larvae in nature [35].

Crushed Japanese elm wood with a relative humidity of 60–70% was used as sawdust. Wetted sawdust was treated in the GC-100-3 steam sterilizer (AO TZMOI, Moscow, Russia) at a temperature of 121 °C for 2 h at a steam pressure of 1 atmosphere. The sterilized sawdust was cooled to a temperature of 25 °C. The treated substrate was transferred to sterile containers and the remaining components of the diet, shown in Table 1, were added.

The prepared containers with all the components of the nutritious diet were incubated in the dark in the MIR-154 cooled incubator (Sanyo, city, Japan) at a temperature of 25 °C and humidity of 70% for about 20 days according to the method developed for *Callipogon relictus* Semenov, 1899 (Coleoptera: Cerambycidae) [36].

The second substrate was the standard diet (SD) used for rearing larvae as food for amphibians and reptiles in insectariums and zoos. SD contains, as a substrate, wheat flakes and bran processed in a microwave oven at a maximum temperature (100 °C) in cycles of 1 min. In addition, once a week, fruit peels, carrots, and other plant residues were added so that the larvae could replenish the moisture.

The prepared substrates were transferred to sterile cages (Ferplast Geo Large, Rome, Italy) 30 cm× 20 cm× 20.3 cm in size, and stock culture eggs were placed there. They were incubated at a temperature of 26–28 °C and 60–70% relative humidity in the MIR-154 cooled incubator (Sanyo, Japan). The surface of the substrate was sprayed with distilled water two or three times a week to avoid drying out. The biomass for the experiments was accrued for almost 2 years (three generations of 7 months each).

To obtain data on larval development, changes in body weight, and consumption of feed substrate, an experiment was conducted where *Z. atratus* larvae of the first instar were kept on FD individually in sterile 100 mL containers (n = 10). Measurements were taken at all stages of the life cycle.

To obtain data on the antibacterial activity of hemolymph *Z. atratus*, larvae of the first instar were kept on FD and SD (without fruit peels, carrots, and other plants) in groups of up to 100 individuals in sterile cages. Larvae were reared until the 12th instar, with an average weight of 756(±)74 mg. Then, 400 pieces were consumed for the experiment (200 pieces for each diet).

### 2.3. Analysis of Hemolymph Antibacterial Activity

Hemolymph was collected with a pipette from the upper abdomen of larvae by needle puncture. Approximately 50 µL of pure hemolymph was isolated from each larva (the quantity varied). Hemocytes were removed by centrifugation on a 5804 R refrigerated centrifuge (Eppendorf, Hamburg, Germany) using an angled rotor for 10 min at 900× *g* with a temperature setting of 4 °C. The resulting plasma was purified by filtration and used for the disk-diffusion test. For filtration, a syringe filter MillexR-MP (LLC Merck, Darmstadt, Germany) with a hydrophilic polyvinylidene fluoride membrane, 10 mm in diameter and a pore size of 0.22 μm, was used. For ultrafiltration, a Microcon YM-30-type filter (LLC Merck, Darmstadt, Germany) with the nominal molecular weight cut-off of 30 kDa was used. Then, the plasma was centrifuged for 1 hour at 10,000× *g* and 4 °C on a 5804 R refrigerated centrifuge (Eppendorf, Germany) with an angled rotor. The fraction below 30 kDa was used in the photometric bacterial test (PBT).

Opportunistic and pathogenic Gram-negative (*Escherichia coli* ATCC 25922, *Pseudomonas aeruginosa* ATCC 9027, *Salmonella abony* 103/39) and Gram-positive (*Bacillus subtilis* ATCC 6633=DSM 347, *Staphylococcus aureus* ATCC 6538—P=GDA 209-P, *Staphylococcus epidermidis* ATCC 14990) bacteria were used as test microorganisms. These strains are used as a control of antibacterial drug efficacy in pharmaceutical production.

Microorganisms were cultured in liquid nutrient medium LB (Appliche, Darmstadt, Germany) (peptone 10 g/L, yeast extract 5 g/L, NaCl 5 g/L, 1 M NaOH 2 mL, pH = 7.0) in an ES-20/60 incubator (Biosan, Riga, Latvia) for a day with constant orbital stirring and temperature regime of 37 °C.

To evaluate the effect of FD on the immunity of *Z. atratus* larvae, the antibacterial activity of isolated hemolymph samples against various strains of bacterial cultures was screened using two tests: disk-diffusion test (DDT) and photometric bacteriostatic test (PBT) were used to evaluate the antibacterial activity. The results were compared with the effect of SD. DDT showed the presence of antibacterial activity in hemolymph and its correlation with feed composition. The FBT showed the bacteriostatic effect of hemolymph during the growth cycle of the test bacteria. The results were compared with the effect of SD.

For the DDT, bacteria of the strains under study were inoculated from the liquid medium using a Drigalsky spatula onto the surface of agarized medium in Petri dishes. Sterile paper disks with a diameter of 0.5 cm were impregnated with plasma samples from larvae reared on FD and SD and placed on fresh bacterial cultures. Four plasma-impregnated disks were placed on one 90 mm diameter dish. The distance from the disk to the edge of the dish and between the disks was 15 to 20 mm. The inoculated dishes were incubated in an ES-20/60 incubator (Biosan, Riga, Latvia) at 37 °C for a day. Antibacterial activity was determined by the size of bacterial growth inhibition zones (with or without single dot colonies) around the sample-impregnated disks. Negative control was sterile disks and positive control was disks impregnated with antibiotic (cefotaxime at a concentration of 0.1 mg/mL). Approximately 2000 μL of purified plasma from larvae reared on the same diet was used for DDT. For PBT, the optical density of daily bacterial culture suspensions was measured in plastic cuvettes with an optical path length of 10 mm using a NanoPhotometer Pearl UV/Vis spectrophotometer (Implen, Munich, Germany). Afterwards, they were diluted in the medium in which they were grown to A600 = 0.05. The optical density of bacteria with samples was measured using a Cytation 5 plate reader (BioTek Instruments, Winooski, VT, USA) with Gen 5 software version 3.08 in “Endpoint” mode (endpoint measurement) at a wavelength of 600 nm every hour for 48 h at an incubation temperature of 37 °C with mechanical agitation before each measurement. Antibacterial activity was determined by the decrease in optical density of bacterial suspensions with samples. Negative control was bacterial suspension and positive control was bacterial suspension with antibiotic (cefotaxime at a concentration of 0.1 mg/mL). Approximately 3000 μL of purified plasma from larvae reared on the same diet was consumed for PBT.

### 2.4. Statistical Analysis

The analysis was performed using Statistica 10 (StatSoft, Tulsa, OK, USA) and GraphPad Prism (GraphPad Holdings, Boston, MA, USA) statistical analysis software version number 8.0.0 packages. The Shapiro–Wilk test was used to test the data for normality of distribution; as a result, nonparametric statistical tests were chosen for further analysis. The Kruskal–Wallis rank-sum test was used to compare 4 independent samples (FD sample, SD sample, positive control, and negative control were compared with each other). The Mann–Whitney U-test was used to evaluate differences between the 2 independent samples (FD sample was compared with SD sample). Standard errors correspond to 95% confidence interval. Results with *p* values < 0.05 were considered statistically significant.

## 3. Results

### 3.1. Effect of FD on Biometric Parameters and Features of Development of Z. atratus Beetles

The obtained biometric indicators and the development duration of larvae of different instars are shown in Table 2.

### 3.2. Effect of FD on the Antibacterial Activity of the Hemolymph of Z. atratus Larvae

The histograms presented in Figure 1 were plotted using the DDT data. A statistical analysis showed significant differences between the compared samples; thus, it can be argued that plasma samples of larvae grown on FD have pronounced bacterial activity as compared to plasma samples of larvae grown on SD. All plasma samples inhibited the growth of Gram-negative bacteria much better than that of Gram-positive bacteria, which was confirmed by the histograms shown in Figure 2. The antimicrobial activity of plasma decreased in the following strains used: *S. abony*, *P. aeruginosa*, *S. epidermidis*, *S. aureus*, *E. coli*, and *B. subtilis.*

PBT measured the optical density for each hour of incubation of bacterial cultures with samples within 48 h. Based on the data obtained, the growth curves of Gram-negative (Figure 3) and Gram-positive bacteria (Figure 4) were constructed. As can be seen from the growth curves, the hemolymph fraction below 30 kDa showed different bacteriostatic properties in relation to the tested bacterial strains. All control cultures of bacteria without additives (negative control) showed a standard growth curve. In cultures with added antibiotic (positive control), no growth was observed; therefore, only negative control is discussed later in the text when describing PBT findings.

An analysis of the growth curves of Gram-negative bacteria showed significant differences in growth under the influence of SD and FD samples as compared to the control. There was virtually no lag phase on the growth curve of the control culture of *E. coli:* during the first hour of incubation, the culture entered the log (exponential) growth phase, which lasted 8 h; then, it entered a long stationary phase. When the SD sample was added, the *E. coli* growth acceleration phase lasted 4 h. The growth curve was similar to the control culture and ran parallel to it, but with a lower growth rate as compared to the control until the end of the experiment. When the FD sample was added, the culture immediately started the log growth that lasted 4 h and was followed by the slowing growth rate (hours 5–7 of the experiment) with a gradual transition to the decline stage; by the end of the second day, the growth was similar to that of the positive control (bacteria with an antibiotic).

There was virtually no lag phase on the growth curve of the *P. aeruginosa* control culture (Figure 3b): during the first hour of incubation, the culture entered the log growth phase, which lasted 11 h. The stationary phase lasted about 13 h, and, on the second day (after 24 h), the culture entered a phase of decline. After the addition of an SD sample, the lag phase lasted for the first 2 h, followed by the log phase for 3 h, stationary phase for 6 h, and, at the 12th hour of the experiment, culture degradation started. When the FD sample was added, the *P. aeruginosa* culture had the same length of phases as when the SD sample was added, but a lower rate, up to the stationary phase, which lasted until the end of the first day (from 6 to 24 h); on the second day, the culture degraded. By the end of the experiment (40 h), the culture started to grow again.

Growth curves of *S. abony* bacteria (Figure 3c) showed the suppression of cultures in the presence of the samples under study as compared to the control. There was virtually no lag phase in the growth curve of the control bacterial culture: in the first hour of incubation, the culture entered the log phase, which lasted 4 h and was followed by a prolonged stationary phase. After the SD sample was added, the growth curve was similar to the control culture and ran almost parallel to it, but at a lower rate until the end of the experiment. The bacterial growth curve in the presence of the FD sample was short (10 h) and then the culture degraded. It is interesting that, after 11 h of cultivation, the bacteria started to grow.

The growth curve of the rod-shaped Gram-positive bacterium *B. subtilis*, shown in Figure 4a, was radically different from the growth curves of the studied Gram-negative bacteria. There was virtually no lag phase on the growth curve of the control culture of *B. subtilis* bacteria: in the first hour of incubation, the culture entered the log phase, which lasted 18 h and was followed by the growth rate slowing down phase, up to hour 26. Then, the culture entered the stationary phase until the end of the experiment. The samples were active only for the first 12 h of incubation; after that, they served as a growth-stimulating nutrient substrate of a sort. Nevertheless, the FD sample was active for a longer time than the SD sample, slowing down the growth of the culture for 2 more hours (from hour 8 to 10), after which a sharp logarithmic growth began.

The growth curves of Gram-positive cocci *S. aureus* and *S. epidermidis* (Figure 4b,c) had similar growth patterns to *P. aeruginosa* and *S. abony*, where, in the initial stages, the growth was suppressed under the influence of SD and FD. There was virtually no lag phase on the growth curve of the control culture of *S. aureus*: in the first hour of incubation, the culture entered the log phase for 8 h, followed by the stationary phase, which lasted 23 h, after which the culture slowly began to degrade. There was also virtually no lag phase on the growth curve of the control culture of *S. epidermidis*: in the first hour of incubation, the culture entered the log phase for 7 h; then, growth began to slow down and, by the beginning of the second day, it reached a plateau (stationary phase). For two cocci cultures, the growth inhibition in the presence of an SD sample lasted for 14 h, followed by a slowly ascending growth curve at the log growth phase and the slowing growth rate phase. Two graphs (for cocci *S. aureus* and *S. epidermidis* in Figure 4b,c) hade an intersection of growth curves of the negative control bacteria and bacteria with an SD sample. For the *S. aureus* culture, this point was at hour 37, and, for the *S. epidermidis* culture, it was at hour 42. For these bacteria, the SD sample served as a growth-stimulating nutrient substrate of a sort. As with Gram-negative bacteria, the FD sample was active for a longer time than the SD sample and slowed down the growth rate of cultures during two days of analysis. Optical density (reflecting bacterial concentration) was almost at the same level.

Similar results are shown in the histograms of viability for 48 h in Figure 5. For Gram-negative bacteria, the samples almost halved the number of bacteria in the suspension. Only for *E. coli*, the SD sample decreased the growth rate by only 35%. The samples had a strong bacteriostatic effect against pathogenic bacteria *P. aeruginosa* and *S. abony*, reducing the bacterial count in the suspension from 50 to 25%. The effect of the samples on Gram-positive bacteria species used was ambiguous. The SD sample suppressed the growth of *S. aureus* and *S. epidermidis* to 53% and 51%, respectively, whereas the FD sample suppressed the growth to 26% and 24%. The opposite was true for *B. subtilis*. With both samples, the bacterial count increased: with FD by 10% and with SD by 50% Figure 6.

## 4. Discussion

*Z. atratus* is an important object of research. Its larvae are saproxylophagous and are of interest to fundamental science as a model object for studying the biology of this insect group. Data on the efficient, economical, and sustainable mass rearing of *Z. atratus* are rather scant, unlike data on other species such as *Tenebrio* (Linnaeus, 1758) (Coleoptera: Tenebrionidae) and *Hermetia illucens* (Linnaeus, 1758) (Diptera: Stratiomyidae), whose industrial rearing is aimed at obtaining not only protein biomass but also valuable secondary metabolites for various purposes.

For the rearing of larvae, wheat bran is usually used with the addition of various fruit and other plant residues; some researchers have used food-industry waste in order to obtain protein biomass of larvae that can be used as an additional source of proteins in the diets of farm animals [37,38]. Information on the mass rearing of beetles on various feed substrates is provided in Table 3.

According to our data, the duration of beetle development from egg to adult on FD is on average 200 days, and the hatching of larvae from eggs usually occurs on day 6–7 after laying; larvae have 12 instars. The duration of development is consistent with the results of a number of the studies presented in Table 2. However, the study by Kim et al. (2015) says that larvae have 18 instars, and it is noted that pupation in some individuals began after moulting to instar 13 (about 3% of the studied cases) [39].

**Table 3 insects-15-00435-t003:** Experimental data on mass rearing of *Z. atratus* larvae in laboratory conditions.

The Diet Used, Specifics of Management	Sources
The main component is wheat bran with the addition of various cereals, such as oat.	[40,41,42,43]
The basic diet based on wheat bran is supplemented with starch-containing products as a source of water (fruit peel, carrots, etc.). If the moisture is insufficient, larvae acquire a cannibal behaviour.	[15,43,44]
Brewery waste.	[44]
Plant residues with the addition of cattle, horse, and chicken manure. Using manure resulted in slowing down of the growth and development rates of larvae.	[45]
Basic diet, temperature 25–28 °C, relative humidity 60–70%. Additionally, the effect of larval density on biomass accumulation was investigated.	[14,15,16,33]

An analysis of plasma antibacterial activity of *Z. atratus* larvae grown on different types of diet showed a selective effect against the studied bacteria. The different effect of plasma samples on Gram-negative and Gram-positive bacteria can be explained by differences in the cell wall structure of prokaryotes. In Gram-positive bacteria, the cell wall is thicker (20–60 nm) and more homogeneous, with a higher content of murein than in Gram-negative bacteria (up to 40 times). From a practical perspective, strong growth suppression (more than 50%) of Gram-negative bacteria by larval plasma is very encouraging. Bactericidal compounds can be divided into three groups according to their principle of action: those destroying the cell membrane, vital prokaryotic proteins, and nucleotides. Regardless of the action, protective substances need to go through the bacterial cell wall, which is thicker in Gram-negative bacteria. The results obtained by us, and confirmed by the literature data, show evidence of an unequalled protection system in higher insects that they developed in the course of long-term evolution.

These protective molecules produced by insects—presumably antimicrobial peptides such as cecropin [46]—seem to have a certain duration of action. In the experiment with *R. aeruginosa*, *S. aureus*, and *S. epidermidis*, FD showed an increase in growth rate after a prolonged bacteriostatic effect. It is believed that the humoral immunity of insects is paraspecific: depending on the nature of the pathogen, peptides of different families [47] acting against certain strains or bacterial groups are synthesized. This statement is consistent with our findings showing that the hemolymph of larvae managed on FD suppresses the growth of all bacteria except *B. subtilis*.

High hemolymph antibacterial activity of larvae grown on FD shows that nutrition is an important factor in immune activity and mediates its relationship with physiological functions. Similar results were obtained with *S. littoralis* larvae. Larvae receiving a high-quality protein diet had higher survival and growth rates than larvae fed a low-quality protein diet; antibacterial activity was higher in insects fed a high-quality diet, which the authors attributed to the general mechanisms of adaptation to the environment in different insect groups [48].

The particular mechanisms of our findings can be interpreted in different ways. Firstly, the humoral mechanisms of the insect immune response include the activation of proteolytic cascades in plasma, the synthesis of antimicrobial peptides being the major one [49,50]. The surface structures of the pathogen are a powerful trigger for the immune response. It is assumed that the biocidal activity of peptides stems from protonated amino groups; the positive charge of the latter causes the biopolymer binding to the anionic components of the microbial cell surface structures (lipopolysaccharides of Gram-negative bacteria) due to electrostatic interaction [51,52]. Therefore, it is possible that, along with the diet, a number of microorganisms got into the insect’s system, which triggered the immune response that we observed in the performed tests.

Secondly, in addition to exogenous pathogenic microorganisms, there is currently conclusive evidence that various nonpathogenic microorganisms can constantly or temporarily colonize the hemolymph in a variety of insects [53,54,55,56,57,58,59]. The hemolymph is a means of transport and the main storage depot for nutrients. Thus, the nutrients obtained from the diet seem to serve as a kind of substrate for the development of microorganisms that are included in the insect microbiome. However, to confirm this hypothesis, microbiome studies are required that are beyond the scope of this work.

The third possible explanation for the high antibacterial activity of the hemolymph of larvae fed the FD is based on the hypothesis of Lee et al. postulating that proteins are responsible for the immune system maintenance or activation [60] and improved nutrition results in fat accumulation and intensive development of the fat body, which is the main source of plasma proteins [24,25,61]. According to this hypothesis, it is perhaps hardly surprising that the larvae fed the standard diet experienced a depletion of internal protein reserves with the concomitant decline in efficiency of the pathogen resistance mechanisms. Because of high metabolic needs associated with the processing of a standard diet consisting mainly of cellulose, energy reserves are limited [62]. On the contrary, the fungi-based diet is rich in proteins that contribute to the improved vitals of larvae (Figure 7).

Presently, vast research has been performed on the effects of diet on the immunity of various insect groups. In these studies, various approaches were used: a reduced amount of food [63], control of the levels of specific nutrients in the diet [64], balancing basic macronutrients [60], and addition of various plant materials [65,66]. From these considerations, we can conclude that the quality of nutrition determines the physiological mechanisms that help the insect to resist infection, thereby promoting good growth and development.

## 5. Conclusions

In the conducted research, FD optimally corresponding to the trophical preferences of the *Z. atratus* beetle in natural conditions was developed and evaluated for cultivation in laboratory conditions. Larvae reared on FD have a shorter development time due to a decreased number of the life cycle stages and their hemolymph shows antibacterial activity against opportunistic and pathogenic bacteria. These data are confirmed by the results of the disk-diffusion and photometric tests. It was found that hemolymph inhibits the growth of Gram-negative bacteria better than that of Gram-positive bacteria.

Thus, the first data on the antibacterial activity of the hemolymph of *Z. atratus* larvae reared in laboratory conditions with different dietary patterns and feed composition were obtained. The results indicate that nutrition determines the immunocompetence of the insect. Given the importance of immunocompetence for detritivores, this work is of fundamental importance to understand the relationship between nutrition and organismal functions in saproxylic Coleoptera and is the first report on immune status regulation in this group of insects. Further research and a search for optimal conditions for mass rearing (production) of insects will provide us not only with new protein sources for farm animal feed but also with broad-spectrum antibacterial compounds and will also help to create lines of saproxylic or saprophagous larvae with the improved immune system and a high level of antimicrobial peptides expression for physiology and ecology studies of Tenebrionid beetles and other insects.

## Figures and Tables

**Figure 1 insects-15-00435-f001:**
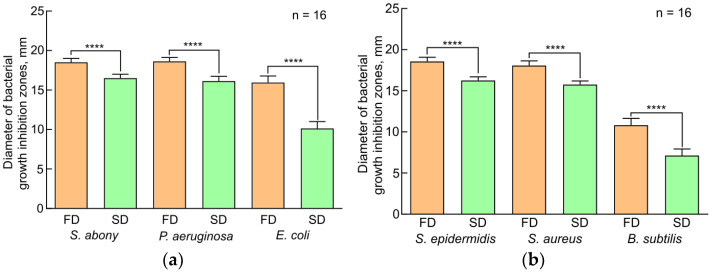
Disk-diffusion test results. The effect of standard diet and fungi-based diet on plasma antibacterial activity of *Z. atratus* larvae, represented as the diameter (mm) of growth inhibition zones of Gram-negative (**a**) and Gram-positive (**b**) bacteria. SD—standard diet, FD—fungi-based diet. The data presented in Figure 1 reflect samples of 16 observations (2 technical re-peats of 8 measurements). Standard errors correspond to 95% confidence interval. Results with *****p* values < 0.05 were considered statistically significant.

**Figure 2 insects-15-00435-f002:**
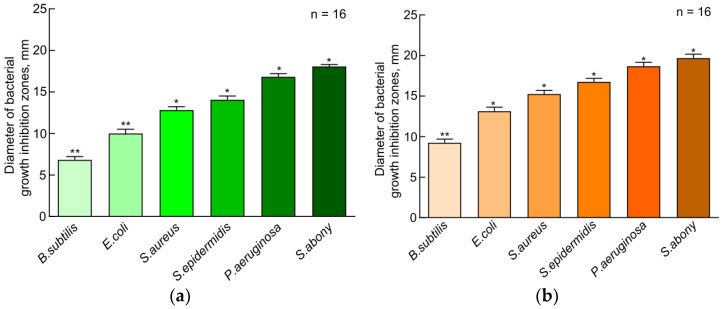
Disk-diffusion test results. Antibacterial effect of hemolymph samples of *Z. atratus* larvae grown on standard diet (**a**) and fungi-based diet (**b**). SD—standard diet, FD—fungi-based diet. The data presented in Figure 2 reflect samples of 16 observations (2 technical re-peats of 8 measurements). Standard errors correspond to 95% confidence interval. Results with all *p* values < 0.05 were considered statistically significant.

**Figure 3 insects-15-00435-f003:**
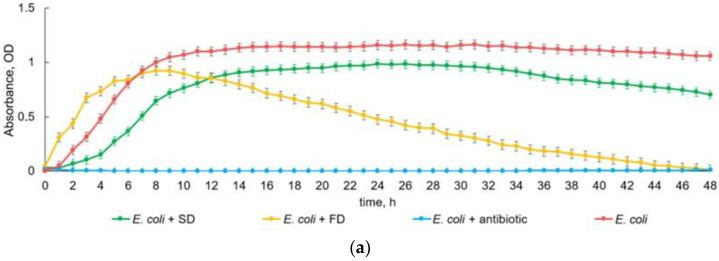
Photometric bacteriostatic test results. The effect of fungi-based diet and standard diet on the antibacterial activity of larval hemolymph of *Z. atratus* within 48 h, represented as the optical density of suspensions of Gram-negative bacteria: *E. coli* (**a**), *P. aeruginosa* (**b**), and *S. abony* (**c**) at a wavelength of 600 nm. The lower the optical density, the fewer bacteria in the suspension. SD—standard diet, FD—fungi-based diet. Each point in the plots reflects the mean value from 8 observations.

**Figure 4 insects-15-00435-f004:**
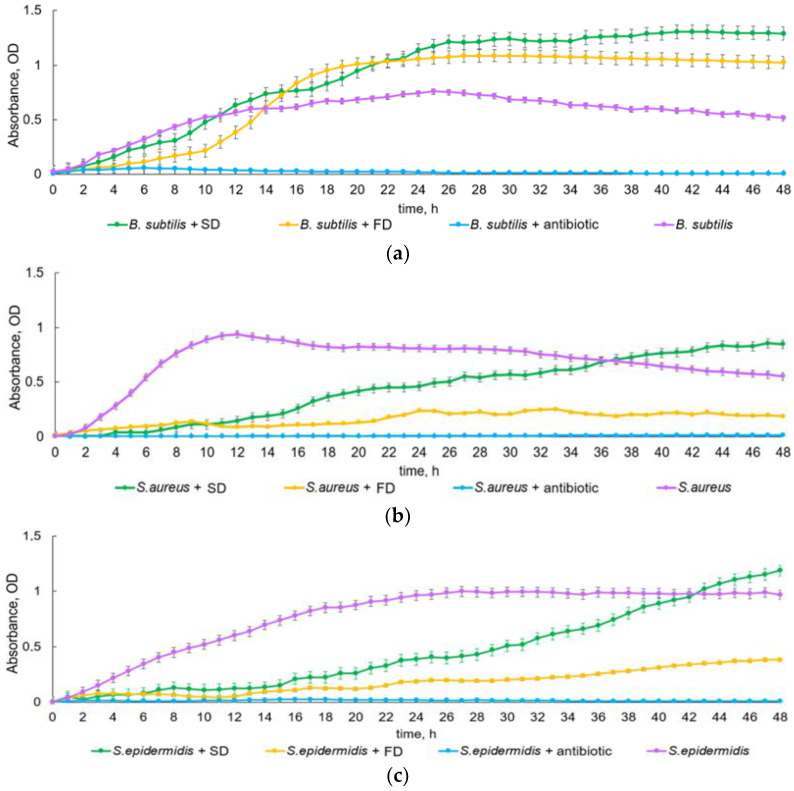
Photometric bacteriostatic test results. The effect of fungi-based diet and standard diet on the antibacterial activity of the hemolymph of *Z. atratus* larvae plotted against time (48 h), represented as the optical density of suspensions of Gram-negative bacteria: *B. subtilis* (**a**), *S. aureus* (**b**), and *S. epidermidis* (**c**) at a wavelength of 600 nm. The lower the optical density, the fewer bacteria in the suspension. SD—standard diet, FD—fungi-based diet. Each point in the plots reflects the mean value from 8 observations.

**Figure 5 insects-15-00435-f005:**
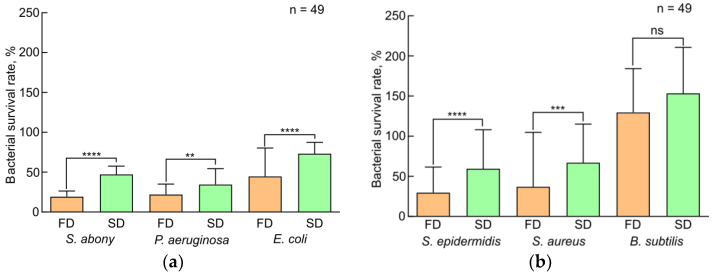
Photometric bacteriostatic test results. The effect of standard diet and fungi-based diet on the antibacterial activity of larval plasma of *Z. atratus*, shown as the percentage of survival of Gram-negative (**a**) and Gram-positive (**b**) bacteria in 48 h. SD—standard diet, FD—fungi-based diet, ns—not significant. The data presented in Figure 5 reflect samples of 49 observations. Standard errors correspond to 95% confidence interval. Results with all *p* values < 0.05 were considered statistically significant; ns—not significant.

**Figure 6 insects-15-00435-f006:**
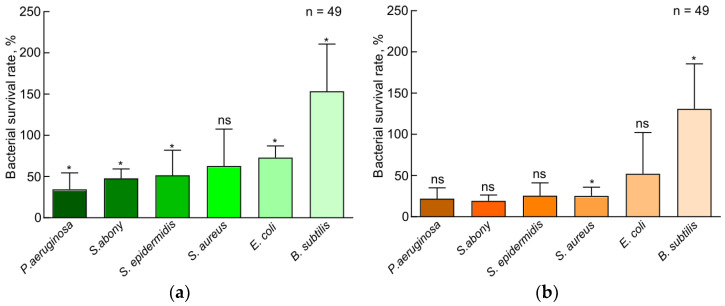
Photometric bacteriostatic test results. Antibacterial effect of hemolymph samples of *Z. atratus* larvae grown on standard diet (**a**) and fungi-based diet (**b**) for 48 h. SD—standard diet, FD—fungi-based diet, ns—not significant. The data presented in Figure 6 reflect samples of 49 observations. Standard errors correspond to 95% confidence interval. Results with **p* values < 0.05 were considered statistically significant; ns—not significant.

**Figure 7 insects-15-00435-f007:**
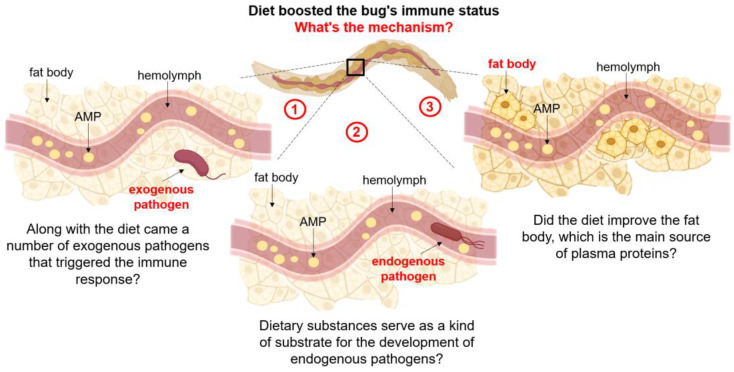
Presumed mechanisms of diet influence on the immune status of *Z. atratus* larvae. 1—Exogenous pathogens entered the organism together with the diet and caused an immune response; 2—diet substances serve as a kind of substrate for the development of endogenous pathogens; 3—the diet improved the state of the fat body, which is the main source of blood plasma proteins. AMPs are antimicrobial peptides.

**Table 1 insects-15-00435-t001:** Components of FD for cultivation of *Z. atratus* in laboratory conditions.

Name of Component	Component Content in 500 g of Medium, g
Sawdust of Japanese elm	120.00
Mycelium of *Pleurotus citrinopileatus*	25.00
Feed yeast	10.00
Ascorbic acid	4.50
Sucrose	20.00
Agar	6.20
Distilled water	314.30

**Table 2 insects-15-00435-t002:** Biometric indicators and duration of development of the *Z. atratus* beetle cultivated on fungi-based diet.

Development Stage	Length, mm	Mass, g	Duration of Development, Days (Mean ± Standard Deviation)
Egg	1.2–1.5	0.1–0.9	6.4 (±) 0.52
Larvae (L1–L4)	2.2–2.7	1.5–3.5	L1—10.8 (±) 0.92
			L2—14.8 (±) 0.79
			L3—12.2 (±) 1.14
			L4—13.6 (±) 1.51
Larvae (L5–L12)	48.0–52.0	10.0–23.0	L5—15.9 (±) 0.57
			L6—15.1 (±) 0.57
			L7—12.6 (±) 1.07
			L8—14.5 (±) 0.71
			L9—12.6 (±) 0.97
			L10—11.4 (±) 0.52
			L11—18.5 (±) 0.97
			L12—23.8 (±) 1.14
Pupa	29.0–45.0	5.6–14.0	9.7 (±) 0.48
Adult beetle	30.0–34.4	–	up to 7 months

## Data Availability

Data are available upon request to the corresponding author of this manuscript.

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
