# Peer review of "Newly Woody Artificial Diet Reveals Antibacterial Activity of Hemolymph in Larvae of *Zophobas atratus* (Fabricius, 1775) (Coleoptera: Tenebrionidae)"

_insects, 2024, doi:10.3390/insects15060435_

Round 1
Reviewer 1 Report
Comments and Suggestions for Authors
Introduction
Lines 45-47
People have used insects as model organisms for a while, not necessarily new. Reword this sentence.
Lines 47-49
Statement ‘insect rearing is cost-efficient’ à not necessarily; it depends on the insects you rear and the type of diet. Reword this sentence.
Materials and Methods
Line 101
Any reason you use the word ‘imago’ instead of adult? The word ‘imago’ is fine but perhaps it should be changed to ‘adult’ for readers who are not familiar with the term. This suggestion applies to other sections of the manuscript.
Lines 102-103
Vladivostok, Primorsky Krai à what country? Please list.
Line 107
Rarely vegetables à should be ‘rare vegetables’. How do you define rare vegetables? Provide examples as well.
Line 112
Option à use different word. Word ‘option’ alluded me that you use either A or B, not both, which is not your case, you used/tested both diets.
Line 147
… 12th instar were selected. à How do you select? Any method use to minimize bias?
Throughout documents:
Spell out FD, SD, PBT or anything abbreviated on figure or table titles to make it clear and for the reason I mentioned next.
In most figures, you have two different SDs: One stands for Standard Diet, and the other stands for Standard Deviation (mean±SD). The latter is commonly used but how do you differentiate two SDs?
Author Response
Dear Reviewer,
Thank you for your time and professional evaluation of the manuscript. We have made changes to the manuscript according to the list below.
Introduction
- Lines 45-47. People have used insects as model organisms for a while, not necessarily new. Reword this sentence.
Revised.
- Lines 47-49. Statement ‘insect rearing is cost-efficient’ à not necessarily; it depends on the insects you rear and the type of diet. Reword this sentence.
Revised.
- Materials and Methods. Line 101. Any reason you use the word ‘imago’ instead of adult? The word ‘imago’ is fine but perhaps it should be changed to ‘adult’ for readers who are not familiar with the term. This suggestion applies to other sections of the manuscript.
Done. Throughout the manuscript, “imago” has been replaced by “adult”.
- Lines 102-103
Vladivostok, Primorsky Krai à what country? Please list.
We added a country.
- Line 107
Rarely vegetables à should be ‘rare vegetables’. How do you define rare vegetables? Provide examples as well.
An error has been made in the spelling in English. We meant that we sometimes add vegetables to the diet, not that these vegetables are rare. Corrected.
- Line 112
Option à use different word. Word ‘option’ alluded me that you use either A or B, not both, which is not your case, you used/tested both diets.
Revised
- Line 147
… 12th instar were selected. à How do you select? Any method use to minimize bias?
Larval stages were determined by molts and head capsule width based Kim, S.Y., Kim, H.G., Song, S.H., & Kim, N.J. Developmental characteristics of Zophobas atratus (Coleoptera: Tenebrionidae) larvae in different instars. International Journal of Industrial Entomology, 2015, 30(2), 45-49, doi:10.7852/IJIE.2015.30.2.45.
https://doi.org/10.7852/ijie.2015.30.2.45
Throughout documents:
Spell out FD, SD, PBT or anything abbreviated on figure or table titles to make it clear and for the reason I mentioned next.
In most figures, you have two different SDs: One stands for Standard Diet, and the other stands for Standard Deviation (mean±SD). The latter is commonly used but how do you differentiate two SDs?
We have supplemented the captions to the drawings with this information. Thank you for pointing out our omission.
Respectfully and gratefully, Authors
Reviewer 2 Report
Comments and Suggestions for Authors
The manuscript is very well written and the subject matter is relevant. I have some minor remarks. Section 3.1 appears to be a repetition of methodology, and could therefore likely be omitted. Secondly, the first paragraphs of the discussion section appears to largely repeat parts from the introduction, and can therefore also be omitted or heavily reduced. Finally, I appreciate that the authors highlight other species such as Tenebrio molitor being increasingly reared commercially, but I think it would be valuable if the authors could circle back to this after discussing the experimental results of this particular study. Although this would only be hypothetical, I think a short discussion of the relevance of the study's results for commercial rearing of other Tenebrionid Beetles.
Author Response
Dear Reviewer,
Thank you for the evaluation of the work.
- We have taken your comments into account. We have deleted section 3.1 and the first paragraphs of the Discussion section.
Finally, I appreciate that the authors highlight other species such as Tenebrio molitor being increasingly reared commercially, but I think it would be valuable if the authors could circle back to this after discussing the experimental results of this particular study. Although this would only be hypothetical, I think a short discussion of the relevance of the study's results for commercial rearing of other Tenebrionid Beetles.
We have made minor corrections to the last paragraph of section "5. Conclusion", which reflected the prospects for studying the physiology and ecology of not only Tenebrionid beetles, but also insects of similar functional groups in general.
Respectfully and gratefully, Authors
Reviewer 3 Report
Comments and Suggestions for Authors
1. Please complete or remove (F.) of the title.
2. Results in abstract were too short, please shorten the introduction and methods.
3. Please merge the first and second paragraph in Introduction.
4. L55-56,“scientists……research” Lack of references.
5. L75“potentially dangerous” What is it? What does it affect the insects?
6. 2.1 Stock culture. How many generations?
7. L131 larvae, What age is the larva?
8. At what development stage do you start breeding insects? To what development stage?
9. L139-140, Is the same added to FD diet? If not, does that cause differences in growth and development, and differences in immunity?
10. Integrate result 3.1 into the materials method.
11. Do you have data on survival rates and reproduction?
12. L213-215, Put it in the last paragraph of the Introduction or material method.
13. L221-223, Put it in the last paragraph of the Introduction or material method.
14. L224, Put it in material method.
15. Figure 1, The first letter of the vertical coordinate should be uppercase. Is n=16 a total sample size for treatment group? What is the relationship with the 400 live larvae mentioned by L147? In the material method, please add the information of each repeat sample size, how many repeats, the total sample size, and so on.
16. nonparametric statistical tests have sample size requirements
17. L235-236, delete it, if it is already mentioned in the material method.
18. The modification suggestions in Figure 2 are the same as those in Figure 1
19. L241-243, Put it in the Introduction or material method.
20. L244-245, Put it in the material method.
21. L287-289, Put it in the material method.
22. n=8, What exactly does that mean?
23. Figure 3, Establish the model. Use the model to illustrate the rule of change, not just describe it.
24. Figure 7?Sequence number error?The modification suggestions are the same as Figure 3.
25. Figure 8,9?
26. L340-341, L344-345, Put it in the material method.
27. Write all the parameters of statistical tests, not just P, H? df?
28. Add references from 2020 to 2024.
Comments on the Quality of English LanguageModerate editing of English language required
Author Response
Dear Reviewer,
Thank you for your time and professional evaluation of the manuscript. We have made changes to the manuscript according to the list below.
- Please complete or remove (F.) of the title.
Executed. We have corrected (F.) to (Fabricius, 1775)
- Results in abstract were too short, please shorten the introduction and methods.
Revised.
- Please merge the first and second paragraph in Introduction.
Revised
- L55-56,“scientists……research” Lack of references.
Thank you for pointing out our omission. We have added references to support the statement we wrote. They are listed under numbers 5-7.
- L75“potentially dangerous” What is it? What does it affect the insects?
We meant by “potentially dangerous” that there are many pathogens in the habitat of xylobionts that affect them. Since the environmental conditions are different for the hard-winged species, we replaced the phrase “potentially dangerous” with “specific environmental conditions”.
- 2.1Stock culture.How many generations?
The eggs were transferred to the cages with diets and three generations were cultivated there.
- L131 larvae, What age is the larva?
First instar larva. We supplemented materials and methods 2.2 with information on larval instars.
- At what development stage do you start breeding insects? To what development stage?
We bought 10 pairs of adult insects for which we created conditions for egg laying. Eggs were transferred to FD and SD diets (we started breeding at the egg stage). Biomass for experiments was accumulated for almost 2 years (three generations of 7 months each). To obtain data on the antibacterial activity of hemolymph Z. atratus larvae of the 1st instar were kept on FD and SD (without fruit peels, carrots and other plant). Twelfth instar larvae were selected for hemolymph isolation. We have revised section 2.2. with this information.
- L139-140, Is the same added to FD diet? If not, does that cause differences in growth and development, and differences in immunity?
Fruit was added when the biomaterial was accrued. For the experiment on antibacterial activity, larvae were cultured without fruit addition, and humidity was compensated by water injection. Section 2.2. was revised and a paragraph on culture propagation was added.
- Integrate result 3.1 into the materials method.
Done
- Do you have data on survival rates and reproduction?
Yes, but we do not report them as part of this study because we are studying the effect of nutrition on the antibacterial activity of hemolymph. The biology data will be published in a separate paper.
- L213-215, Put it in the last paragraph of the Introduction or material method.
Put it in the material and method
- L221-223, Put it in the last paragraph of the Introduction or material method.
Done
- L224, Put it in material method.
Done
- Figure 1, The first letter of the vertical coordinate should be uppercase. Is n=16 a total sample size for treatment group? What is the relationship with the 400 live larvae mentioned by L147? In the material method, please add the information of each repeat sample size, how many repeats, the total sample size, and so on.
Figures have been edited according to the observation.
n=16 - so many observations (measurements) in one sample in DDT.
Approximately 400 pc larvae were consumed for the antibacterial activity experiment. Of these, 200 pc were kept on SD diet and 200 pc on FD diet. Approximately 50µl of pure hemolymph was isolated from 1 larvae (this number varied from individual to individual). From 200 larvae, approximately 10000µl of hemolymph was isolated. After the plasma was separated and purified, the volume was halved. 5000µl of the experimental sample (purified plasma) was used in antibacterial tests. About 2000µl was used for DDT (it took so much to impregnate 16 disks with one sample), 3000µl for FBT. Supplemented this information with materials and methods.
- nonparametric statistical tests have sample size requirements
For the Mann-Whitney test: 1) each sample must contain at least 3 observations: n1,n2 ≥ Z; it is allowed that one sample contains 2 observations, but then the second sample must contain at least 5 observations; 2) each sample must contain no more than 60 observations; n1, n2 ≤ 60.
For the Kraskell-Wallis test: there must be at least 5 observations in each sample (I have not found restrictions on no more than some number).
We have for DDT-16, and for FBT-49 observations in each sample. We are within the sample size requirements of these statistical tests.
- L235-236, delete it, if it is already mentioned in the material method.
Executed.
- The modification suggestions in Figure 2 are the same as those in Figure 1
Executed.
- L241-243, Put it in the Introduction or material method.
Executed.
- L244-245, Put it in the material method.
Executed
- L287-289, Put it in the material method.
Executed
- n=8, What exactly does that mean?
Number of observations (measurements), the average of which corresponds to each point on the graph (Fig. 3,4) / number of wells of a 96-well plate allocated per sample in which optical density was measured every hour for 48 hours in the FBT method. Added this information to Materials and Methods.
- Figure 3, Establish the model. Use the model to illustrate the rule of change, not just describe it.
Done
- Figure 7?Sequence number error? The modification suggestions are the same as Figure 3.
The figure has been assigned the number 4. Thank you for pointing out our omission.
- Figure 8,9?
The figures have been assigned numbers 5 and 6. Thank you for pointing out our omission.
- L340-341, L344-345, Put it in the material method.
Executed
- Write all the parameters of statistical tests, not just P, H? df?
The analysis was performed using Statistica 10 (StatSoft, USA) and GraphPad Prism 8 (GraphPad Holdings, USA) statistical analysis software packages. The Shapiro-Wilk test was used to test the data for normality of distribution, and as a result, nonparametric statistical tests were chosen for further analysis. Kruskal-Wallis rank-sum test was used to compare 4 independent samples (FD sample, SD sample, positive control, and negative control were compared with each other). The Mann-Whitney U-test was used to evaluate differences between the 2 independent samples (FD sample was compared with SD sample). The data presented in Figs. 1, 2 reflect samples of 16 observations (2 technical repeats of 8 measurements), those in Figs. 5, 6, reflect samples of 49 observations. Each point in the plots (Figs. 3, 4) reflects the mean value from 8 observations. Standard errors correspond to 95% confidence interval. Results with p values < 0.05 were considered statistically significant. We attach the file Fig.1a with DDT results (Effect of standard diet and mushroom-based diet on antibacterial activity of blood plasma of Z. atratus larvae represented as diameter (mm) of zones of growth inhibition of Gram-negative bacteria) as an example of the reliability of our results.
- Add references from 2020 to 2024.
New references have been added. They are listed under the numbers: 6, 7, 34, 38.
Respectfully and gratefully, Authors
